

# Absorption of VOCs by polymer tubing: implications for indoor air and use as a simple gas-phase volatility separation technique

Melissa A. Morris[1], Demetrios Pagonis[1]*, Douglas A. Day[1], Joost A. de Gouw[1], Paul J. Ziemann[1], Jose L. Jimenez[1]

[1]Department of Chemistry and Cooperative Institute for Research in Environmental Sciences (CIRES), University of Colorado, Boulder, CO, 80303 USA
* Now at: Department of Chemistry and Biochemistry, Weber State University, Ogden, Utah, 84408 USA

*Correspondence to*: Jose L. Jimenez (jose.jimenez@colorado.edu)

**Abstract.** Previous studies have demonstrated volatility-dependent absorption of gas-phase volatile organic compounds (VOCs) to Teflon and other polymers. Polymer-VOC interactions are relevant for atmospheric chemistry sampling, as gas-wall partitioning in polymer tubing can cause delays and biases during measurements. They are also relevant to the study of indoor chemistry, where polymer-based materials are abundant (e.g. carpets, paints). In this work, we quantify the absorptive capacities of multiple tubing materials, including 4 non-conductive polymers (important for gas sampling and indoor air quality), as well as 4 electrically-conductive polymers and 2 commercial steel coatings (for gas and particle sampling). We compare their performance to previously-characterized materials. To quantify the absorptive capacities, we expose the tubing to a series of ketones in the volatility range $10^4$-$10^9$ $\mu g\ m^{-3}$ and monitor transmission. For slow-diffusion polymers (e.g. PFA Teflon and nylon), absorption is limited to a thin surface layer, and a single layer absorption model can fit the data well. For fast-diffusion polymers (e.g. polyethylene and conductive silicone) a larger depth of the polymer is available for diffusion, and a multilayer absorption model was needed. The multilayer model allows fitting solid phase diffusion coefficients for different materials, which range from 4 x $10^{-9}$ to 4 x $10^{-7}$ $cm^2\ s^{-1}$. These diffusion coefficients are ~8 orders of magnitude larger than literature values for FEP Teflon. This enormous difference explains the differences in VOC absorption measured here. We fit an equivalent absorptive mass ($C_w$, $\mu g\ m^{-3}$) for each absorptive material. We found PFA to be the least absorptive, with $C_w$ ~$10^5$ $\mu g\ m^{-3}$, and conductive silicone to be the most absorptive, with $C_w$ ~$10^{13}$ $\mu g\ m^{-3}$. PFA transmits VOCs easily, and intermediate-volatility species (IVOCs) with quantifiable delays. In contrast, conductive silicone tubing transmits only the most volatile VOCs, denuding all lower volatility species. Semi-volatile species (SVOCs) are very difficult to sample quantitatively through any tubing material. We demonstrate how to use a combination of slow- and fast-diffusion tubing to separate a mixture of VOCs into volatility classes before analysis. New conductive silicone tubing contaminated the gas stream with siloxanes, but this effect was reduced by 10,000-fold for aged tubing, while maintaining the same absorptive properties. SilcoNert (tested in this work) and Silonite (tested in previous work) steel coatings showed gas transmission that was almost as good as PFA, but since they undergo adsorption, their delays times may be humidity and concentration dependent.



## 1 Introduction

It is well known that gas-phase volatile organic compounds (VOCs) partition in and out of polymers by absorption.
Absorption and diffusion of organic species through polymers has been studied since the 1940s (Crank and Henry, 1949) due to practical applications such as polymer dye efficiency (Chantrey and Rattee, 1974) and permeation of aromas through food packaging (Johansson and Leufven, 1997).

Polymers compose a large fraction of surfaces in indoor environments, including painted surfaces, carpets and synthetic fabrics. Manuja et al. (2019) measured total surface area for 10 bedrooms, 9 kitchens and 3 office spaces in the
U.S. On average, the percent of surface area attributed to carpets and fabrics ranged from 2% (kitchens) to 22% (bedrooms). Paint was the largest contributor overall, averaging 40% of total surface area across all room types (see Table S1). VOCs found indoors are lost primarily through ventilation and deposition, and can be emitted or re-emitted from polymers (Pagonis et al., 2019; Price et al., 2019). Understanding the extent, dynamics, and mechanism for polymer absorption of VOCs is therefore important for understanding and better quantifying the loss processes for VOCs in indoor environments. Most past
indoor studies quantified the first-order deposition rate constant for VOCs to a surface or film, but did not discuss the absorption mechanism or diffusion into the bulk of the material. Algrim et al. (2020) demonstrated the extremely high absorptive capacity of paint films, as VOCs diffused through the entire thickness of the paint film in a matter of hours, making paint a likely major sink and reservoir for VOCs in indoor environments. Won et al. (2000) found that synthetic carpets take up large quantities of VOCs when concentrations are high, and re-emit VOCs when concentrations are low,
acting as large indoor reservoirs of VOCs. Synthetic carpets, therefore, are expected to increase the indoor persistence of VOCs through reversible partitioning. However, the partitioning of VOC to carpets has not been studied with a level of detail that may allow predictive modeling, to our knowledge.

Polymer tubing is used extensively to sample air and gases for studies of indoor air and outdoor atmospheric chemistry, due to properties such as inertness, flexibility, and low-cost. However, if the gas contains VOCs, interactions
between those compounds and the tubing walls can lead to delays and detection biases. Several recent studies (Pagonis et al., 2017; Deming et al., 2019; Liu et al., 2019) quantified the VOC adsorptive properties of perfluoroalkoxy alkane Teflon (PFA), fluorinated ethylene propylene Teflon (FEP), and other polymer tubings. These studies grew from the literature on vapor losses to Teflon walls of environmental chambers (Matsunaga and Ziemann, 2010; Krechmer et al., 2016; Huang et al., 2018). Absorption of VOCs by polymer tubing is chromatographic in nature, reversible, and independent of compound
concentration or humidity (Pagonis et al., 2017; Deming et al., 2019). Absorption varies strongly with the polymer material, with e.g. the same VOC species resulting in 1 minute of delay per meter of PFA for a given setup, and 10 minutes of delay per meter of polytetrafluoroethylene Teflon (PTFE) in the same setup (Deming et al., 2019). Delays are also a strong function of saturation mass concentration ($C^*$, i.e. vapor pressure in mass units) with an order of magnitude more delay for an order of magnitude decrease in $C^*$ or flow rate (Pagonis et al., 2017).





A single tubing layer, one-dimensional, mass flow model was developed to describe VOC absorption by PFA and other less-sorbing polymers, where diffusion into the bulk of the polymer is assumed to be much slower compared to gas flow down the length of the tubing (Pagonis et al., 2017). Using the model, experimental partitioning delays can be fit to find an equivalent absorbing mass, or absorptive capacity, of the tubing walls ($C_w$, in µg m$^{-3}$). For more absorptive polymers, like paint, where diffusion into the bulk of the polymer cannot be assumed to be much slower than gas flow down the length of

the tubing, a multilayer absorption model was needed to capture the experimental features (Algrim et al., 2020).

       Non-polymer tubes (including metal and glass tubes) induce longer partitioning delays than PFA. Unlike polymers, the partitioning delays through metal and glass tubes depend on competition for surface sites between the species sampled, and thus concentration, humidity, and exposure history affect the transmission times (Deming et al., 2019).

       Most previous studies of tubing-wall interactions thus far have been aimed at increasing the accuracy of

measurements, by reducing and/or accounting for VOCs lost or delayed by the sampling setup. However, this phenomenon may also have some practical applications. For example, polymer tubing was recently used to separate isomers and identify fragments of the same species in PTR-MS (Jenks et al., 2023). Additionally, the tubing geometry provides major advantages over larger reactors, chambers, or rooms for studying material-VOC interactions. Tubing studies have high experimental reproducibility due to repeated opportunities for contact with the walls and the decreased influence of convection and mixing

compared to larger spaces (Krechmer et al., 2016; Pagonis et al., 2017).

       In this work, we experimentally characterize VOC absorption in polymer materials and commercial steel coatings, and compare our results with previous literature. The polymers we test and model in this work are significantly more sorptive than previously studied materials, and the application and further development of a multi-layer sorption model was necessary to explain observations and determine their sorptive capacities. A subset of these materials (nylon, polyester,

polypropylene, and polyethylene) are highly relevant to indoor air quality, as they are prevalent polymers that compose synthetic carpet. We investigate both electrically conductive and non-conductive polymer tubing, which are used for gas and particle sampling, and gas-only sampling, respectively. Data are fit with the single and multilayer absorption models to extract relevant parameters for each tubing material. Finally, we demonstrate how a given polymer tubing, characterized for VOC absorption, can be used to selectively transmit higher-volatility species (a "volatility high-pass filter"), and how a

combination of tubing materials can allow characterization of VOC mixtures by volatility ranges.

## 2 Methods

### 2.1 Measurements

#### 2.1.1 Measurements of partitioning delays

To test polymer tubing for VOC sorption, we employ the same methodology as previous studies (Pagonis et al., 2017;

Deming et al., 2019; Liu et al., 2019). A schematic of the sampling setup is shown in Fig. 1A. In short, a large (either 20 or 7



$m^3$) FEP chamber was used to provide a stable reservoir of VOCs, and the tubing of interest was installed between the chamber and the detector. Measurements of $C_6$-$C_{12}$ 2-ketones through 7 different tubing materials were conducted; the relevant details for the compounds and materials are listed in Table S2. The chamber contained approximately 20 ppb of each ketone. This was prepared by injecting known amounts of liquid ketones into a sealed glass bulb, attaching the bulb to

the chamber with a flow of nitrogen (~5 L min$^{-1}$), and gently heating the outside of the bulb to evaporate and transfer the ketones. The chamber was left to equilibrate for at least 30 minutes prior to experiments, so that gas-wall partitioning was complete (Matsunaga and Ziemann, 2010; Krechmer et al., 2016). A Vocus proton-transfer-reaction mass spectrometer (hereafter referred to as "Vocus") (Krechmer et al., 2018) was used to monitor ketone concentrations at 1 Hz time resolution. The Vocus was connected to the chamber using short (<1 m) PFA lines and fittings. Flow from the chamber to the Vocus

was controlled using a critical orifice to 2 L min$^{-1}$. The tubing materials were tested for VOC absorption by installing one at a time between the PFA lines in the sample path. With the tubing of interest installed, first the chamber was sampled for up to an hour or until concentrations plateaued, and then room air was sampled until the concentrations returned to levels that were similar to instrument backgrounds.

         We define the partitioning delay time as the time it takes for the tubing output from the start of desorption to reach
10% of the maximum concentration during the absorption period, as is done in previous studies (Pagonis et al., 2017; Deming et al., 2019; Liu et al., 2019). The instrument and the short PFA lines cause short partitioning delays, which were subtracted from the total delay time, to quantify just the delay time associated with the tubing being tested. Delay times for the setup were on the order of 0.01-1 min m$^{-1}$, and were significantly smaller than for the polymers studied, except for nylon, conductive PFA, and conductive PTFE. Given the time resolution of the system, the shortest quantifiable delays were of ~1

second. For longer desorption periods, the time series were fit with a single exponential between 5-15% of the maximum concentration, and the 10% time was calculated using the fit, reducing the effect of noise. For the few desorption periods that were very long (~hours) and did not reach 10% of the maximum concentration during the absorption period, the desorption periods were fit with double exponentials, and the delay times were extrapolated out to the 10% concentration. The extrapolated delay times are consistent with trends measured in lower capacity materials. All final tubing delay times were

normalized by dividing by the tubing length.

**2.1.2 Measurements of partitioning delays with the gas volatility separator**

The gas volatility separator (GVS) is a multi-tube autosampler. A schematic is shown in Fig. 1B. The GVS is designed to flow a sample gas stream through multiple tubing materials simultaneously, to separate VOCs of different volatility ranges. In the design used here, a common inlet is split into four flow paths, each containing one tubing material. Upstream of each

inlet tube is an automated PFA 3-way solenoid valve (Cole-Parmer, Part No. EW-01540-18), which allows each tube to ingest either a sample gas stream or clean air. Downstream of each tubing material, another automated PFA 3-way solenoid valve directs the flow to either the common sampling outlet or a vacuum pump exhaust (with critical orifices to match sampling flow rates). This configuration ensures that the VOC analytes enter all tubing materials at a constant flow rate,





regardless of which inlet path is being sampled by the Vocus. The valves are wired to a solid state relay board, and are
operated by the control and acquisition software (MICAS, Original Code, Boulder, CO) within Labview. Figure S1 shows
the custom user interface. The user can manually switch which tubing material stream is sent to the outlet, or set up a
software sequence to cycle through all of the materials with a set dwell time for each. The GVS can switch between tubing
materials as fast as every 30 seconds, but we found that switching tubing materials every 3 minutes gives more interpretable
results with the tube lengths used in this study, as time is required for the PFA lines (after the outflow of all the tubing
materials join at the same point) to re-passivate after switching. The common outlet can be sampled by an instrument
directly. The whole system is mounted on a custom aluminum rack (80-20, https://8020.net/), so that it can be easily
transported for field studies. Photos of this system are shown in Fig. S2. For this work, the inlet of the GVS was connected to
a chamber containing 2-ketones as in section 2.1, and the outlet was connected to the Vocus, using a 4 m PFA line. The four
tubing materials installed for the tests described here were PFA, polypropylene, polyethylene and conductive silicone
(lengths of 1.2 - 4.5 m, see Table S2).

## 2.2 Modeling

### 2.2.1 Single layer model for slow-diffusion materials

A single-layer absorption numerical model developed and published by Pagonis, et. al. (2017) (freely available at:
https://tinyurl.com/PartitioningDelays) was used to quantify the absorptive capacities of slow-diffusion polymers, and
provide a rough estimate of those of fast-diffusion polymers. A detailed description of the model is available in that
publication (Pagonis et al., 2017). In short, the model is a linear kinetic chromatography model that calculates gas to wall
partitioning, as compounds move down the length of the tube with the gas flow, according to Eq. (1):

$$F_W = \frac{1}{1 + \frac{C^*}{C_W}} \qquad (1)$$

where $F_w$ is the fraction of compound in the tubing wall at equilibrium, $C^*$ (μg m$^{-3}$) is the saturation vapor pressure of the
compound, and $C_w$ (μg m$^{-3}$) is the equivalent absorbing mass, or absorptive capacity, of the tubing material per unit volume
of air in the tube. Values for $C^*$ were estimated using the SIMPOL.1 group contribution method (Pankow and Asher, 2008).
Molar mass of the partitioning phase (polymer) is assumed to be 250 g mol$^{-1}$, with an activity coefficient of 1. This molar
mass is typical of organic compounds in atmospheric aerosol particles, where activity coefficients are usually assumed to be
1, and is used in place of the polymer molecular weights, which are typically unknown (Algrim et al., 2020). The gas-phase
diffusion coefficient used for all ketones was 0.067 cm$^2$ s$^{-1}$, which was estimated for dodecanone (Tucker and Nelken, 1982).
This value was not changed for each individual ketone, as it has little effect on the delay time (Pagonis et al., 2017). The
model assumes diffusion into the bulk of the tubing material is very slow, so absorption occurs in a thin, homogeneous, near-
surface layer. For example, for PFA tubing the thickness of that layer is estimated to be about 2.2 nm (Krechmer et al., 2016;



Pagonis et al., 2017). The model output is a time series of compound concentration exiting the tube, which can be compared to experimental data.

We fit a $C_w$ value to each slow-diffusion material by minimizing the sum squared residual between the experimental time series and modeled time series, for all ketones simultaneously. For fast-diffusion materials, we employed the method of

165 Pagonis, et al. (2017) and Deming, et al. (2019): experimental delays for all ketones were plotted in log-log space as a function of $C*$ (as shown in Fig. 5), the model was run with a range of $C_w$ values, modeled delays as a function of $C*$ were calculated, and the $C_w$ value that provided delays closest to the experimental ones (based on an orthogonal distance regression) was chosen.

### 2.2.2 Multilayer model for fast-diffusion materials

A multilayer absorption-diffusion numerical model was first developed and published to model absorption of VOCs by thick (few hundred μm) paint films (Algrim et al., 2020). This model was used to fit the fast-diffusion materials in this study with $C_w$ values and solid phase VOC diffusion coefficients, for each ketone. In short, the model shares the same gas-to-wall partitioning framework as the single layer model, but also includes diffusion of the compounds into the bulk of the polymer, according to Fick's second law of diffusion:

$$\frac{\partial C(x,t)}{\partial t} = D_f \ \frac{\partial^2 C}{\partial x^2} \tag{2}$$

where $D_f$ is the solid phase diffusion coefficient, and $C_{(x,t)}$ is the concentration at depth $x$, and time $t$. The equation is used in Cartesian (instead of cylindrical) coordinates, since the diffusion depth is small compared to the tubing diameter (< 10%, and

180 much less for most materials), and given other uncertainties in the model. Equation 2 is incorporated in the model as a finite difference approximation with the same Euler timestep as the gas-surface layer partitioning calculation. As in the previous case, the model output is a time series that can be compared to experimental data.

Changes were made to the Algrim multilayer model for use in this work. The Algrim multilayer model fits a VOC-paint system with 3 parameters: a solid phase diffusion coefficient ($D_f$, cm² s⁻¹), a surface roughness factor ($R$, unitless), and

185 an absorptive capacity ($C_w$, μg m⁻³). The surface roughness factor scales the surface absorption rate coefficient, to simulate turbulence at the gas-polymer interface, and allows the amount of surface layer absorption at short times to be adjusted. The updated model fits a VOC-polymer system with a solid phase diffusion coefficient ($D_f$, cm² s⁻¹), a surface roughness factor ($R$, unitless), and a mass fraction availability parameter ($Y$, unitless). Instead of fitting $C_w$, the updated model calculates $C_w$ based on the modeled partitioning depth (which is a function of $D_f$) and the density of the polymer. The mass fraction

availability parameter is a new fitting parameter that scales the now-calculated $C_w$ value by the fraction of the polymer mass available for gas-wall partitioning. Without $Y$, the model could not reproduce the experimental time series; consistent overestimation of absorption occurred when it was assumed that all the mass in the surface layer of the polymer was



available for partitioning. The mass fraction parameter is mathematically equivalent to the inverse of an activity coefficient (resulting from the chemical interaction between the absorbing species and the polymer). The same result could be obtained by using a larger value for the polymer molecular weight (vs. the constant value of 250 g mol$^{-1}$ used here). E.g. $Y = 0.03$ is mathematically equivalent to an effective polymer molecular weight of ~8000 g mol$^{-1}$, which is physically plausible. A possible physical interpretation is that larger polymer chains, potentially with more cross-linking, may leave less space for the absorbing species to occupy. Figure S3 compares the fit values for $Y$ from this work with activity coefficients from the literature. From the currently available data it remains unclear whether this effect is the result of chemical activity, physical exclusion, or some combination of the two. This topic should be explored further in future studies.

The multilayer model fits 3 parameters ($D_f$, $R$, $Y$) for each ketone-polymer system. $D_f$ was allowed to vary with each ketone, but $R$ and $Y$ were held constant for each polymer, as they are more empirically related to the material than to individual VOCs. To do this, we carried out a brute-force fitting routine for each of the 3 parameters, running the model for each combination of parameters, and then calculating the sum squared residual between the experimental and modeled time series. This created a 3D cube of fit errors, which was manually evaluated for each ketone and also across ketones. This process was done iteratively at an increasingly fine scale until we converged on a set of parameters that best modeled all the ketone time series for a given polymer. Calculated $C_w$ values were averaged across ketones after fitting. Often, the smaller ketones were less sensitive to changes in the $R$ and $Y$ than the larger ones, so it was useful to start by fitting the time series of the larger ketones, then fit the smaller ones, and then iterate.

## 3. Results and discussion

### 3.1 Characterizing tubes as either fast- or slow-diffusion polymers

### 3.1.1 Experimental time series

Figure 2 shows the experimental time series of $C_6$-$C_{14}$ ketones sampled through several different tubing materials. If the tubing had not absorbed any VOCs, these traces would be square waves; instead they show partitioning delays according to the volatility of the compounds and the absorptive capacity of the material. Figure S4 shows a simplified version of the ketone transmission through these polymers using a "stop light" categorization. While materials like PFA and nylon transmitted the ketones with measurable delays, polypropylene and polyethylene demonstrated continued absorption over the 20 minute sampling period, and the $C_{14}$ ketone was lost to the walls in both tubes ($C_{12}$ and $C_{13}$ were also lost to the walls of the polyethylene tube).

### 3.1.2 Fitting absorptive capacities and diffusion coefficients

The single layer model was used to fit all of the materials with an absorptive capacity. The single layer model can reproduce time series through slow-diffusion materials very well. While the single layer model can reproduce partitioning delays as



large as those seen through the fast-diffusion materials, when the modeled time series and experimental time series are compared, it is clear that the single layer model is not capturing the true behavior of the absorption. In panel A of Fig. 3, the modeled and experimental time series for nylon agree fairly well, and the multilayer model is not necessary. In panel B of Fig. 3, the single layer model matches the experimental partitioning desorption delay time for polypropylene, but does not capture the continued absorption seen in the time series, so the multilayer model was necessary. The multilayer model was able to reproduce time series through fast-diffusion materials fairly well. Figure S5 shows the final model fits compared to the experimental time series for all ketone-polymer systems. Any tubes that required the multilayer model were deemed fast-diffusion polymers. Table 1 summarizes all tubes as fast-diffusion, slow-diffusion or adsorptive materials.

A summary of the fit parameters and calculated partitioning depths are summarized in Table 2, and more details are included in Table S3. Since $C_w$ is dependent on the geometry of the tubing, we included dimensions in Table S2, so that $C_w$ values fit here can be scaled for other surface-to-volume ratios (Pagonis et al., 2017). Table 2 also includes an estimated partitioning depth for each of the materials. To estimate the partitioning depths for slow diffusion polymers, we employ the method of Krechmer et al. (2016), where $C_w$ and polymer density are used to estimate partitioning depth. To estimate the partitioning depths for fast-diffusion polymers, we employ Eq. (6) from Algrim et al. (2020), which uses the diffusion coefficient and timescale to estimate partitioning depth. Figure S6 shows good agreement between estimated partitioning depths and multilayer model simulations; details are in the caption.

Unlike the single layer model, the multilayer model fits a diffusion coefficient for each VOC in each tubing material, in addition to the absorptive capacity. Figure 4 compares the fit diffusion coefficients in this work to literature values for other polymers. Our fit values are consistent with literature values for polymers that are not FEP. The $D_f$ estimates for FEP included in Fig. 4 are 8 orders-of-magnitude lower than for the other polymers, and show a similar $C^*$ dependence at the higher $C^*$ values. This enormous difference in diffusion coefficients is consistent with how we can easily categorize materials as either slow-diffusion or fast-diffusion polymers. Notably, FEP tubing is often used as a permeation tube for VOCs. Due to its slow-diffusion properties, VOCs will permeate its walls at a steady rate over the course of several months. Our tests are performed over 10-90 minute periods, so VOC diffusion into slow-diffusion materials is effectively confined to a very thin surface layer.

### 3.1.3 Modeling complexities of fast-diffusion materials

While the multilayer model was able to reproduce the experimental time series for fast-diffusion materials fairly well, it was limited in its ability to capture the different absorption behavior shown by these materials. The desorption of the fast-diffusion materials proceeded at a changing rate, initially being faster than the corresponding absorption rate, and later being slower than the corresponding absorption rate (as shown in Fig. S7). This meant that, at short times, compounds diffused out of the polymer faster than they went in, and at long times, compounds diffused out of the polymer slower than they went in. The behavior at long times is expected; to enter the polymer, compounds just partition to the surface, but to exit the polymer, compounds must diffuse back to the surface before re-partitioning into the gas phase, therefore taking longer to exit than



enter. The behavior at short times was a surprise. However, this anomaly has been published in polymer absorption literature from the 1950s, and remains a complex phenomenon, with factors like non-Fickian solid phase diffusion coefficients, polymer relaxation and rearrangement rates, or changing surface concentrations cited as possible explanations (Crank, 1951; Crank and Park, 1951). We noticed this anomaly in our data while modeling; the multilayer model could reproduce either the absorption period or the desorption period for a material very well, but had to split the difference between the two periods to model the whole time series (as shown in Fig. S8). We tried to account for this anomaly by modeling a depth-dependence and a concentration-dependence for the solid phase diffusion coefficient, but none of the attempted dependences recreated the experimental time series better than the original model. At the expense of an improved fit, we opted to keep the model simple, as it fits the data fairly well, and so that it can remain applicable with good accuracy to the wide range of materials found in indoor environments and sampling tubes.

**3.2 Summary of materials for gas transmission and separation**

The time it took each ketone to return to 10% of the maximum concentration is shown for each material in Fig. 5. As expected from prior literature results, the delays caused by partitioning for a given tubing material increase with decreasing saturation vapor pressure. The relative trend is very similar for different materials, despite several orders of magnitude differences in response times.

The desorption delay times are used to create Fig. 5, instead of the absorption delay times, because the fast-diffusion materials plateaued at different concentrations depending on their absorptive capacities. It is worth noting that for slow-diffusion materials, the absorption and desorption delay times are not identical; often the absorption delay times are larger than the desorption times (Fig. S9a and S9b). There was also more run-to-run variability in the absorption times than desorption times (see Fig. S9c). Using the desorption delay times to summarize this work does not affect the fit values, because the entire time series are used for fitting.

**3.3 Summary of materials for gas and particle transmission and separation**

**3.3.1 Conductive polymers**

In many applications, transmission of both gases and particles is critical. A substantial fraction of ambient particles are electrically charged. Those particles are lost very quickly when sampled through non-conductive materials such as PFA, as surface charges build up that attract the particles, leading to electrophoretic losses (McMurry and Rader, 1985). For this reason, particle sampling is often accomplished with metal tubes such as steel or copper. On the other hand, sampling of gases with minimum perturbation of the chemical composition requires inert polymer tubings such as PFA, because uncoated metal tubes can produce large and unpredictable partitioning delays (Deming et al., 2019). Previous studies concluded that aluminum-foil wrapped conductive PFA (cPFA) was the best known tubing for transmission of both phases (Liu et al., 2019). In this work, we did not come across any tubing that performs better than aluminum-foil wrapped cPFA.



However, there are multiple other conductive tubing materials, which may be of interest for applications such as volatility separation of gases.

In Figure 6A, 2-ketones are sampled through several conductive polymers: cPFA, conductive polytetrafluoroethylene (cPTFE), conductive polyurethane (cPUN) and conductive silicone (cSI), as well as non-conductive PFA. The cPTFA and cPTFE tubes gave similar delays, although slightly longer, than non-conductive PFA, while the cPUN and cSI tubing denuded the ketones majorly. The cSI tubing completely denuded all but the 2 most volatile ketones, transmitting $C_6$ and $C_8$ at ~80% and ~30% of their respective chamber values, after an hour of sampling. cSI has trouble transmitting species with $C^* < 10^7$ µg m$^{-3}$.

Since cPUN and cSI appeared potentially useful for denuding gases according to volatility, we investigated the release of impurities from these materials. It has been previously reported (Timko et al., 2009; Yu et al., 2009; Asbach et al., 2016) that conductive silicone tubing emits siloxanes, particularly polydimethylsiloxane (PDMS). Figure 6B shows the mass spectra of gas phase emissions from cSI tubing (new and >5 year-old pieces of tubing) and cPUN tubing (new). The mass spectra confirm the presence of siloxanes when sampling through cSI, as previously reported in the literature. To our knowledge, VOC emissions from cPUN tubing have not been reported before, so suspected peaks are labeled with mass to charge ratios. There is no significant difference between the VOC absorption of the new and old cSI tubing, indicating that age does not affect the absorptive capacity of the tubing, consistent with previous results for an FEP chamber (Matsunaga and Ziemann, 2010). However, contaminant emissions are significantly reduced with age. Fig. 6B demonstrates a factor of 10,000 decrease in emissions between the new and old cSI tubing. This suggests the impact of siloxane contaminant emissions on measurements using old cSI tubing may be significantly reduced compared to new tubes. The emissions from cPUN tubing are 2 orders-of-magnitude lower than for cSI tubing, but may still cause significant interference in VOC measurements.

Figure 5 shows a summary of the partitioning delays through all the materials in this study that can be used for gas and particle transmission. Of note is how much more absorptive cSI tubing is than any other material we have tested, causing partitioning delays that are 2-3 orders of magnitude larger.

**3.3.2 Stainless steel coatings**

In addition to the polymer tubes discussed so far, three stainless steel tubes were tested for VOC partitioning delays, one uncoated and two with commercial passivation coatings (SilcoNert® and Dursan®, from SilcoTek®, https://www.silcotek.com/). The time series when sampling a step function of the ketone mixture through the steel tubes is shown in Fig. 7. The coated tubes caused partitioning delays that were an order of magnitude larger than through PFA, with Silconert performing better than Dursan. As discussed previously, VOC delays through metal tubes are dependent on humidity, concentration, and history. The desorptions for these tubes were carried out with humid air (~40% RH), while the ketone sampling was performed with dry air. The desorption process can proceed faster than the absorption process for an adsorptive material exposed to water vapor, as water can outcompete less polar compounds (like the ketones) for adsorptive



sites. This effect is not apparent for the SilcoNert coated tube, but it is clear for the Dursan-coated tube and the bare stainless steel. The spike at the beginning of the uncoated stainless steel tube is an artifact of competitive adsorption, and consistent with prior observations (Deming et al., 2019). In comparison, the PFA absorption and desorption timescales are unaffected by changes in humidity (Deming et al., 2019).

The coated tubes were also tested for particle transmission. Ambient particles were pulled through the tubing to a
condensation particle counter, and there was no significant difference between the particle transmission for the uncoated steel tube and coated tubings, as shown in Fig. S10. Some losses are apparent for all tubes compared to the room, and they are tentatively attributed to the loss of ultrafine particles (often observed in room air at this location) via diffusion to the tubing walls.

The partitioning delays through these materials are compared to those of conductive polymers in Fig. 5. The
absorption models were not used to fit the results from these materials, as only adsorption of compounds to surface sites plays a role for metal tubing (Deming et al., 2019).

One possible application of conductive commercial coatings is to potentially increase gas transmission inside atmospheric chemistry reactors. For example, oxidation flow reactors (OFRs) are typically built with chromated aluminum interiors, which has been shown to cause significant losses of gases via humidity dependent adsorption (Deming et al.,
2019). In the supplemental, we compare gas and particle transmission through a chromated aluminum OFR with an OFR that had been coated with a commercial conductive PTFE-based coating, and discuss when it may be useful to use a coated one.

## 3.4 Using the GVS for gas phase separation

The gas-volatility separator (GVS) was used to sample $C_4$-$C_{16}$ ketones with four different polymer tubes. All four tubes sampled the ketones continuously, but only the output of one of the tubes was measured at a given time, which was
alternated every 60 s. An example time series for this experiment is shown in Fig. 8A. The top of the graph indicates which tubing is being sampled at a given time; the experiment starts with sampling through PFA, which was the least absorptive tubing, and then samples through polypropylene, polyethylene and finally cSI, which was the most absorptive tubing, before cycling again to PFA. As time progresses, all the ketone signals increase at different rates through each tube. When a PFA sampling period follows a cSI sampling period, the ketone concentrations rise quickly before reaching their expected
transmission levels. This is because the cSI tubing is so absorptive that the shared PFA lines downstream of the GVS are depassivated while the cSI tubing is sampled, and when the system switches over to PFA from cSI, those lines reabsorb the ketones similar to the start of the experiment. This effect can be minimized by sampling each tube for a longer period of time (e.g. 3 minutes) and only recording concentrations at the end of the 3 minute period. This experiment clearly demonstrates the large differences in transmission of VOCs between polymers. These results can be expressed as a fractional transmission
vs. $C^*$ for a given material and sampling time. Fig. 8B shows the fractional transmission curves for the experiment in Fig. 8A, after 10 minutes of sampling.





Figure 8B shows the model-measurement comparison for PFA and cSI, while Fig. S11 shows the rest of the polymers. The PFA model runs were convolved with the Vocus instrument response (Liu et al., 2019) to better represent the experimental data. The model results compare well with the experimental data, except for the higher volatility ketones and PFA, where experimental delays are larger than predicted. We believe this due to the other components of the setup (e.g. Teflon 3-way valves and fittings) causing more sorption than the components included in the model (tubing and instrument only).

Two modeled fraction transmission curves for PFA are also shown, after 1 minute and at 10 minutes of sampling, to illustrate how these curves change with sampling time. Fig. 8B makes clear that PFA transmits VOCs easily, and intermediate-volatility species (IVOCs) with quantifiable delays. In contrast, conductive silicone tubing transmits only the most volatile VOCs, denuding all lower volatility species. Semi-volatile species (SVOCs) are very difficult to sample quantitatively through any tubing material we have tested, since none perform better than PFA.

Analogous to gas chromatography, there are several experimental variables that affect the transmission curves, including the $C^*$ of the VOC, the $C_w$ of the tubing material, the length and diameter of the tubing, temperature, and flow rate (Pagonis et al., 2017). Some of these can be adjusted to optimize the setup for a particular sampling application. Analysis of data from such experiments is complicated by the time-dependence of the transmission curves. However, the single and multilayer models can be used to model different setups. A grid of delay times as a function of $C^*$ of the VOC and $C_w$ of the transmitting material is shown in Fig. S12, which can act as a guide for separation of gases using polymer tubes.

There are many applications where separation of gasses by volatility class with polymer tubing would be useful. One demonstration was published by Jenks, et al. (2023), who use PFA tubing to identify parent-fragment ion relationships in the time series of measurements from a chemical ionization mass spectrometer, where otherwise two different reaction products may have been assigned. This was achieved by flowing reaction products from a Δ-carene oxidation experiment through a 50 m coil of PFA tubing, and identifying signals at different $m/z$ that had the same desorption profiles through the tubing, suggesting a parent-fragment ion relationship.

Another application of this technique would involve sampling air into an oxidation flow reactor (OFR) through one or several conductive polymer tubes. An OFR can be used to quantify the amount of secondary aerosol mass formed from gas-phase precursors (Ortega et al., 2016; Kang et al., 2007). When put in front of an OFR, conductive polymer tubes act as volatility high-pass filters, only transmitting higher volatility VOCs, while also transmitting particles. This would allow one to quantify the amount of aerosol formed from species above a certain volatility cutoff. Measurements of total OH or ozone reactivity could be made according to volatility class in a similar manner.

## 4. Conclusions

Polymer absorption of VOCs is known to be dependent on compound volatility and the absorptive capacity of the polymer. In this work, the absorptive capacities of many polymers, relevant for new air sampling techniques and indoor air quality,



were quantified. Experimental time series were fit with either a single layer or multilayer absorption model, and polymers were categorized as either slow-diffusion or fast-diffusion. Both of the models used in this study are freely available to download at https://tinyurl.com/PartitioningDelays. VOCs penetrated only a surface layer < 1 μm (often only tens of nm) thick for slow diffusion polymers during our experiments. In contrast, fast-diffusion polymers took compounds into the bulk of the material on a timescale of minutes, with penetration depths of 10-100 μm after 20 minutes. This finding has significant implications for indoor air, as much of the surfaces in indoor environments are made of polymers, and it is usually assumed that sorption only happens in a thin near-surface layer, or simply just on the surface of materials. Nylon, polyester, polypropylene and polyethylene are the main polymers found in synthetic carpets, and quantifying their absorptive capacities adds to the growing literature that indoor polymers can provide a huge sorptive reservoir for VOCs.

Materials in this study were also categorized by application: non-conductive polymers for gas transmission; and conductive polymers and coated metal tubes for gas and particle transmission. The best material for gas transmission was PFA Teflon tubing, and the best material for gas and particle transmission was foil-wrapped conductive PFA; both of these findings are consistent with previous tubing literature. The most sorptive material was cSI tubing, which did not transmit species with $C^* < 10^7 \mu g \ m^{-3}$. The cSI tubing (as well as the cPUN tubing, which also showed significant denuding), could be used to purposefully denude species of lower volatilities from a sample gas stream. However, cSI contaminates the gas stream with siloxanes, which is problematic for sampling applications (cPUN tubing also emits unknown compounds in significant amounts). Older cSI tubing offgassed significantly less contamination, and it may be possible to accelerate such aging (e.g. with controlled heating in an inert atmosphere). SilcoNert (tested in this work) and Silonite (tested in previous work) performed almost as well as PFA in terms of gas transmission. However, since these are steel coatings they undergo adsorption, which makes their delay times potentially dependent on humidity, concentration and history. These effects are very difficult to predict or correct for in the real-world, so non-polymer tubing is only recommended when gas-wall interactions are not important (e.g. when sampling dust or metal particles, or very volatile gases).

Additionally, using a series of polymer tubes with different absorptive capacities, we demonstrate how partitioning delays caused by polymer tubing can be exploited as a volatility class-separation technique for atmospheric chemistry sampling. This technique can be utilized in future studies to measure volatility dependence of compound characteristics like secondary organic aerosol SOA formation potential, or radical reactivity.

**Code and data availability**

The models used in this work are freely available at https://tinyurl.com/PartitioningDelays.



**Author contributions**

MM and JJ designed the experiments. MM performed the measurements, analyzed the data and wrote the manuscript draft. DP and DD assisted with performing the measurements. JJ and DP assisted with measurement interpretation and modeling.
All authors provided input during review meetings along the project, and reviewed and edited the draft manuscript.

**Competing interests**

The authors declare that they have no conflict of interest.

**Acknowledgements**

We thank the Alfred P. Sloan Foundation (grant no. 2019-12444), NSF AGS 2206655, and the CIRES Innovative Research
Program for funding this study. We also thank Jesse Bischof of SilcoTek for supplying the coatings and for useful discussions, Andy Lambe of Aerodyne for supplying a coated OFR from Aerodyne Inc. and for useful discussions, and Pedro Campuzano-Jost and Anne Handschy for help with laboratory work.

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



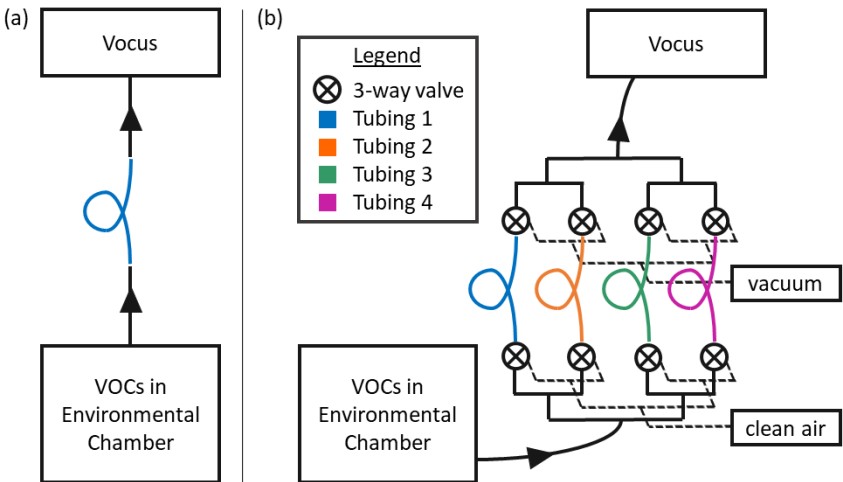

**Figure 1. Schematics for the two sampling setups used in this study. (a, left): Setup for testing the absorptive capacity of one polymer tubing at a time. (b, right): Schematic of the gas volatility separator (GVS). In both cases, VOCs are injected into a large FEP chamber, allowed to equilibrate, and then sampled through the polymer tubing with a Vocus-PTR-mass spectrometer. Solid black lines represent PFA tubing; all unions and valves were also made of PFA.**

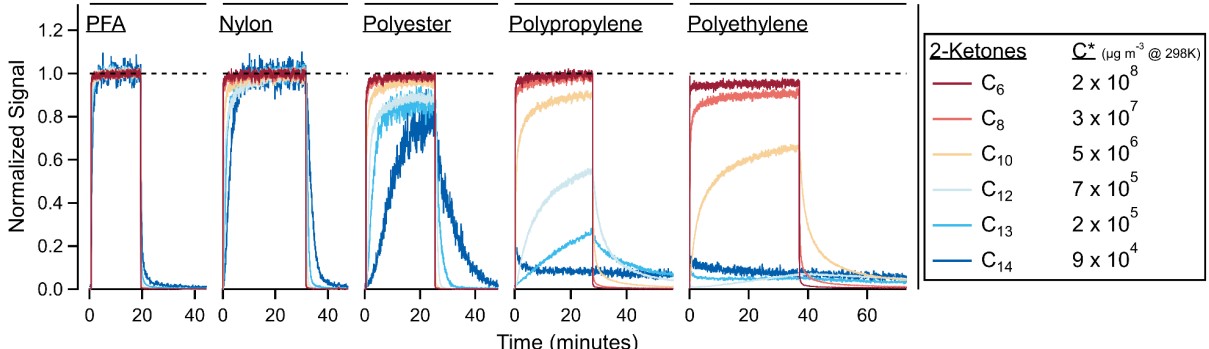

**Figure 2. Time series for 2-ketones of different carbon numbers through a variety of tubing materials. In the absence of tubing absorption, these time series would be square waves. Interactions between the tubing wall and the ketones cause partitioning delays. Materials are ordered from least to most absorptive: PFA (5.8 m length), nylon (1.5 m), polyester (1.8 m), polypropylene (4.6 m), and polyethylene (2.5 m). The $C_{13}$ and $C_{14}$ traces were smoothed by ~5 seconds to reduce noise.**





**Figure 3. Model runs with the single layer model (SLM) and multilayer model (MLM), evaluated against experimental time series of ketones through two materials. A) For a slow-diffusion polymer like nylon, the single layer model is adequate for fitting, although not perfect (see text). B) For a fast-diffusion polymer like polypropylene, the multilayer model is required, as the single-layer model performs poorly. The lack of symmetry in the experimental absorption and desorption periods (for both types of polymers) is discussed below (sect. 3.1.3).**

**Table 1. Categorization of polymer-containing materials tested in this study and in other literature (\*^)**

| Diffusion Regime | Used for gas transmission | Used for gas and particle transmission |
|---|---|---|
| **Slow-diffusion** | PFA, PTFE, Nylon, FEP*, PEEK* | cPFA, cPTFE |
| **Fast-diffusion** | Polypropylene, Polyethylene | cSI, cPUN |
| **Adsorption only** | | SilcoNert, Dursan, Silonite^ |

*(Deming et al., 2019)

^(Liu et al., 2019)



**Table 2. Fitted model parameters for absorption of 2-ketones in polymer tubing materials from this study and the literature.**

| Tubing material | Average mass absorptivity, $C_w$ ($\mu g\ m^{-3}$) | $\text{Log}_{10}$ diff. coeff. for $C_6$ ketone ($cm^2\ s^{-1}$) | Mass fraction availability parameter (unitless) | Surface roughness parameter (unitless) | Estimated diffusion depth at 20 mins for $C_6$ ketone ($\mu m$) |
|---|---|---|---|---|---|
| *Perfluoroalkoxy alkane | $8.0 \times 10^5$ | - | - | - | 0.0018 |
| *cPFA | $1.3 \times 10^6$ | - | - | - | 0.0029 |
| *Fluorinated ethylene propylene | $2.0 \times 10^6$ | - | - | - | 0.0043 |
| *Polyether ether ketone | $8.0 \times 10^6$ | - | - | - | 0.023 |
| *Polytetrafluoroethylene | $1.2 \times 10^7$ | - | - | - | 0.026 |
| *cPTFE | $1.6 \times 10^7$ | - | - | - | 0.035 |
| Polyester | $2.4 \times 10^7$ | - | - | 1.3 | 0.11 |
| cPFA (cond. perfluoroalkoxy alkane) | $2.5 \times 10^7$ | - | - | 1 | 0.055 |
| Nylon | $3.5 \times 10^7$ | - | - | 1.4 | 0.098 |
| cPTFE (cond. polytetrafluoroethylene) | $9.3 \times 10^7$ | - | - | 1 | 0.20 |
| Polypropylene | $3.0 \times 10^{10}$ | -7.4 | 0.02 | 2.9 | 110 |
| Polyethylene | $3.0 \times 10^{11}$ | -6.4 | 0.03 | 2.75 | 340 |
| cSI (conductive silicone) | $6.9 \times 10^{13}$ | -6.5 | 1 | 9.2 | 310 |

*Results for $C_w$ from Deming, et. al. 2019. Depths calculated here from $C_w$ results.

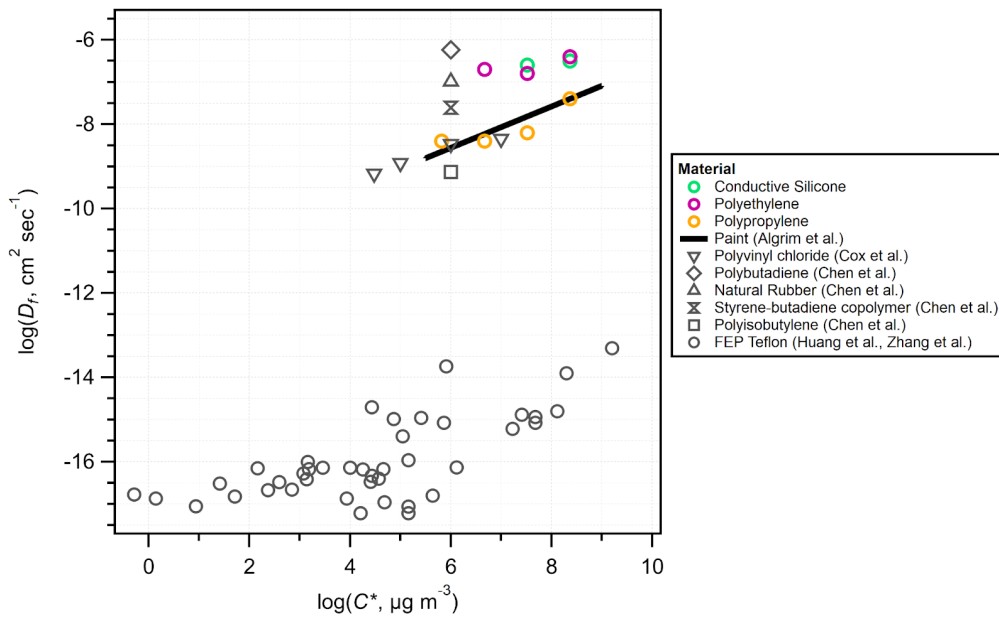

**Figure 4. Log-log plot of diffusion coefficients of VOCs in polymers against VOC saturation mass concentrations values (estimated using SIMPOL.1 for 298 K). Data from this study is plotted alongside literature values with consistent results. Of note is the ~8 orders of magnitude difference between diffusion coefficients for the fast-diffusion polymers and FEP.**



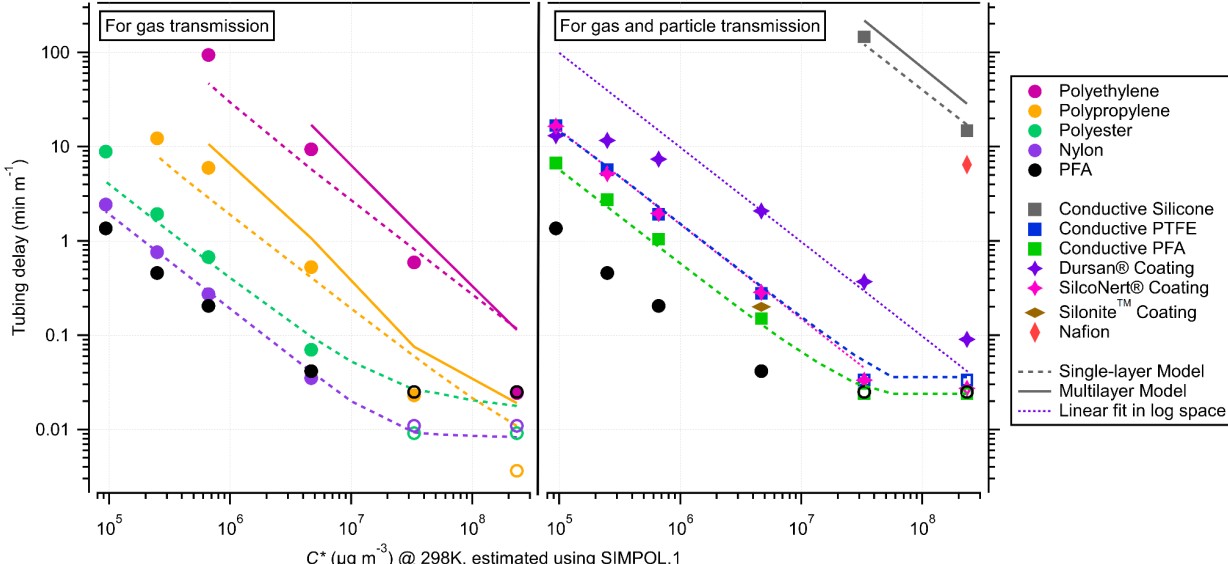

**Figure 5. Experimental partitioning delays in different tubing materials (measured at a flow rate of 1.86 L min⁻¹) vs. saturation mass concentration for 2-ketones. On the left are polymer tubes used for sampling gases, and on the right are conductive polymer tubes and coated metal tubes used for sampling gases and particles. Single layer model results are shown as dashed lines and multilayer model runs are shown as solid lines (for the fast-diffusion polymers). The coated metal tubes were not modeled, but fit linearly in log space with a slope that matched the single layer model runs of other materials, to guide the eye for comparison. Non-conductive PFA is shown on both sides to aid comparison, even though it does not transmit particles well. Data points with open markers indicate that delays may be smaller than the 1 second measurement limitation. One data point from Deming, et al. for Silonite™ tubing is shown for comparison; this data point was taken at a flow rate of ~0.3 LPM, so it may be biased high. Deming, et al. report Silonite™ tubing to be similar in delay times to PFA when tested at the same flow rate. Additionally, one data point from Deming, et. al. for Nafion is included here for comparison, which has been scaled for flow rate.**





**Figure 6. A)** Time series of 2-ketones sampled as step functions through different conductive tubing materials, cPFA, cPTFE, cPUN and cSI, compared to standard non-conductive PFA. The $C_{13}$ and $C_{14}$ traces were smoothed by ~5 s to reduce noise. **B)** Vocus mass spectra of the suspected emissions from new cSI tubing, >5 years-old cSI tubing, and new cPUN tubing. Both old and
555 new cSI tubing emit siloxanes (labeled with elemental formulas from the literature), however the new tubing emits ~10,000 times more than the old tubing. The inset graph shows the scale of emissions from the old tubing. The cPUN tubing emits VOCs at concentrations 2 orders of magnitude lower than the new cSI tubing.



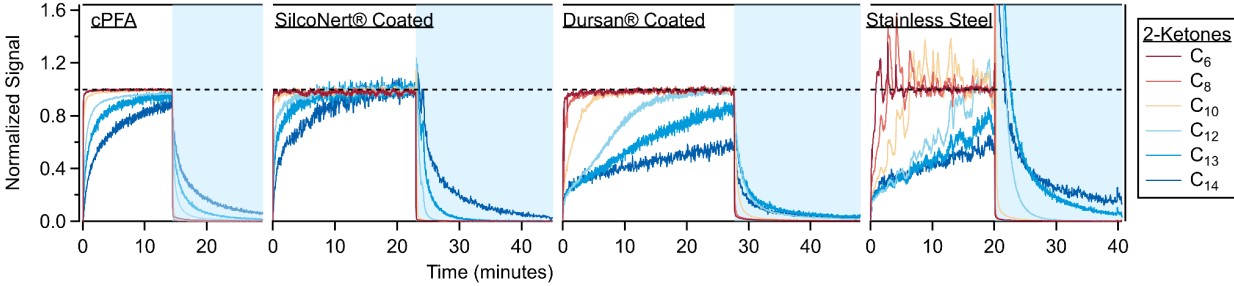

**Figure 7. Time series when sampling step functions of 2-ketones through cPFA tubing (149 cm) and 3 stainless steel tubes (61 cm). One tube was coated with SilcoNert® 2000, the second with Dursan®, and the third was left uncoated. The $C_{13}$ and $C_{14}$ ketone traces for both panels were smoothed by ~5 s to reduce noise. The sampled air from the chamber (periods with white background) was dry (<1% RH), while the room air sampled during desorption periods (blue background) contained ~40% relative humidity. Transmission through metal tubes is typically affected by humidity; as water competes for adsorption sites (Deming et al., 2019). The spike in the stainless steel time series continues up to a value of 10.**





**Figure 8. A) Time series of C₄-C₁₆ ketones through the gas volatility separator (GVS). The GVS inlet tubing was automatically switched every minute, and the inlet tubing material is labeled above the data. B) Cityscape lines show experimental fraction transmission after about 10 minutes of sampling, according to C\*, for various polymer tubing materials. Smooth lines show model runs for cSI after 10 minutes and PFA after 1 and 10 minutes. Model runs for all other polymers are consistent with experimental data, and are shown in Fig. S11. See text for a discussion of the PFA modeling. Compound names have been added to the top of the graph for reference to relevant C\* values.**