# Peer review of "Absorption of VOCs by polymer tubing: implications for indoor air and use as a simple gas-phase volatility separation technique"

_EGUsphere, 2023_

## Author Comment (AC1)

**Response to reviewers for the paper "Absorption of VOCs by polymer tubing: implications for indoor air and use as a simple gas-phase volatility separation technique" Melissa A. Morris , Demetrios Pagonis , Douglas A. Day , Joost A. de Gouw , Paul J. Ziemann , and Jose L. Jimenez**

We thank the reviewers for their comments on our paper. To guide the review process we have copied the reviewer comments in black text. Our responses are in regular blue font. We have responded to all the referee comments and made alterations to our paper **in bold text**. All line numbers refer to the originally submitted manuscript.

**Anonymous Referee #1**

Overview:
R1.0. In article EGUSPHERE-2023-1241, "Absorption of VOCs by polymer tubing: implications for indoor air and use as a simple gas-phase volatility separation technique," the authors present an in-depth discussion and model of uptake of gases into tubing materials widely used by the atmospheric measurement community, as well as a technique and apparatus for exploiting this process to provide potential resolution for direct-sampling instruments such as chemical ionization mass spectrometers. This work is a sort of continuation of a number of useful and important papers on this topic from this research group and advances the ideas by providing a model for use. It is generally very well written and presents the data clearly and usefully. I have a few core questions that I would like to see the authors respond to, but I believe only relatively minor revisions are necessary prior to publication.

Major comments:
R1.1. In this and previous work, the authors use only a series of ketones. I would like to see a mention or discussion of what impact this might have on the results. Do the authors believe there would be any effect due to e.g., polar functional groups or other chemical classes?

The authors thank the reviewer for this very important point. The following text has been added to line 97:

**"In previous works (Matsunaga and Ziemann, 2010; Liu et al., 2019), n-alkanes, 1-alkenes, 2-alcohols, 2-ketones, hydroxynitrates, dihydroxynitrates, and dihydroxycarbonyls were tested for absorption in Teflon tubing. The relative trend in absorption was well described by the vapor pressure estimates of these compounds, with an apparent activity coefficient effect for more polar compounds (Krechmer et al. 2016; Liu et al. 2019; Huang et al. 2018). Since polarity and functional group effects were reasonably understood from previous works, we decided to use a series of 2-ketones to test the materials in this study."**

In addition, the following text was added to line 270:

 **"PFA continues to be the best tubing material for fast gas transmission. One interesting material to note is nylon, which has very short delay times, similar to those of PFA. While nylon is able to transmit the 2-ketones very quickly, it is not an inert material like PFA, and thus is less suitable for**

**general-purpose sampling. Due to the basic amide in its monomer unit, nylon wool is highly efficient at scrubbing acids from a gas stream (Huey et al., 1998)."**

R1.2(a) It is not completely clear how the MLM and SLM are related, for instance in Sections 2.2.1 and 2.2.2. Are they wholly different? It seems to me that the multi-layer model should collapse to the single layer model for the case that the fraction of polymer available for bulk uptake goes to essentially 0, is that the case? Presumably there is at least some capability for bulk uptake even in the single-layer materials.

Yes, the MLM is built on top of the SLM framework. The authors have made a few edits to clarify this.

Lines 170-174 originally read:

"A multilayer absorption-diffusion numerical model was first developed and published to model absorption of VOCs by thick (few hundred μm) paint films (Algrim et al., 2020). This model was used to fit the fast-diffusion materials in this study with Cw values and solid phase VOC diffusion coefficients, for each ketone. In short, the model shares the same gas-to-wall partitioning framework as the single layer model, but also includes diffusion of the compounds into the bulk of the polymer, according to Fick's second law of diffusion…"

and have been replaced with:

**"A multilayer absorption-diffusion numerical model was developed on top of the single layer model to simulate absorption of VOCs by thick (few hundred μm) paint films (Algrim et al., 2020). In short, the model shares the same gas flow and gas-to-wall partitioning framework as the single layer model, but also includes diffusion of the compounds into the bulk of the polymer, according to Fick's second law of diffusion…"**

And the following text was added to line 181:

**"When run without diffusion into the bulk, the multilayer model collapses onto the single layer model."**

We agree that the single layer materials have some bulk uptake. However, our estimates show that the materials modeled well by the SLM have a partitioning depth of ~10 nm. This depth is easily modeled with a "single layer" of material, where we do not need to include simulation of the diffusion process. This vastly decreases the computational run time and the complexity of the model.

R1.2(b) Is there a downside to using the multi-layer model for all tubing? Does it better represent the time series of the slow diffusing materials (e.g., nylon, Figure 3)?

Theoretically, the MLM could be used to simulate all tubing materials. However, there were two practical challenges we came across when trying to do this in our work. First, the MLM requires fitting 3 parameters, while the SLM is a 2-parameter fit. In our experiments, slow-diffusion materials typically depassivated in less than a minute. When using the MLM to fit the slow-diffusion materials, we found

that multiple combinations of the 3 parameters could recreate the time series, without fitting it better than the SLM. Since the MLM was overfitting the data, we could not confidently assign a diffusion coefficient and $C_w$ value. Conversely, with the longer desorption timescales of the fast-diffusion materials, there was a clear solution for the 3 parameters that best simulated the data. Second, the SLM is easier to use (it has a user-friendly panel interface) and has a run time that is at least an order of magnitude faster than the MLM. Although the MLM is optimized to the best of our coding abilities, single runs can take up to a couple hours depending on the tubing material and simulated amount of time. During the fitting process, we input a grid of values, doing hundreds of runs to find optimized parameter values, so always relying on the MLM could become burdensome for some applications.

R1.2(c) If so, what qualifies as "requir[ing] the multilayer model" to be deemed fast-diffusion (line 230).

We tried modeling all the materials with the SLM first, and if the behavior in the experimental time series could not be recreated with the SLM, then the material "required" the MLM, and was deemed to have "fast-diffusion". When we calculated the normalized fit errors for all the model runs, it was apparent that model runs with a normalized fit error of 0.01 (normalized to the length of the model runs and the maximum y-value) or larger required the MLM. We have compiled a table of normalized errors and included it in the SI (reproduced below). We have made the following changes to the text in order to make this evaluation more quantitative.

Lines 229-230 originally read:

"Any tubes that required the multilayer model were deemed fast diffusion polymers."

and have been replaced with:

**"We defined the normalized fit error as:**

$$Normalized\ Error\ = \frac{\sum(C_{experimental}(t) - C_{model}(t))^2}{n} \tag{3}$$

**where $C_{experimental}(t)$ is the experimental concentration (normalized to the chamber value) at time $t$, $C_{model}(t)$ is the normalized concentration predicted by the model at time $t$, and $n$ is the number of data points. Any materials that had a normalized fit error larger than 0.01 with the SLM showed significant deviations from the experimental data in the time series. They were deemed fast diffusion polymers, and were modeled with the MLM instead. A list of normalized fit errors is included in Table S4."**

The following table was added to page 3 of the supplemental:

**Table S4. Normalized Fit Errors for Comparison**

| Tube Material & 2-Ketone | SLM Normalized $X^2$ Error | MLM Normalized $X^2$ Error |
|---|---|---|

| Nylon C14 | 0.0028 | not needed |
|-----------|--------|------------|
| Polyester C14 | 0.0083 | not needed |
| Polypropylene C12 | 0.12 | 0.0049 |
| Polyethylene C10 | 0.38 | 0.0024 |
| Conductive silicone C8 | 0.055 | 0.00015 |

R1.2(d) Similarly, if the MLM were used for slow diffusers, couldn't you extract the partitioning depth directly and directly compare Df between slow and fast diffusers, instead of relying solely on the literature values in Figure 4 (lines 239-247)?

Theoretically yes, the MLM could be used to extract a diffusion coefficient from the slow-diffusion materials as well. However, as discussed in R1.2(b), the MLM tended to over-fit the data we had for the slow-diffusion materials, and could not unequivocally constrain the diffusion coefficient. In our experimental work we used a consistent flow rate of ~2 LPM to conduct all the tubing absorption tests. If other researchers were interested in using the MLM to fit the slow-diffusion materials, we would recommend using a much lower flow rate and try fitting that dataset. However, this may still not be successful, since the uptake into the bulk is 8 orders of magnitude slower than for the fast-diffusion materials, and therefore this technique may simply not be sensitive enough to changes in concentration over time to quantify the diffusion coefficient properly.

R1.2(e) Also, I'm not totally clear, how many layers are in the multi-layer model? Is it just a surface and a bulk, or are there varying depths? Or am I just thinking about it wrong?

The following text was added to line 181 to clarify this point:

**"Each layer in the multilayer model is 2 µm thick, and the number of layers simulated depends on the expected partitioning depth, calculated from the input diffusion coefficient and run time. The model simulates enough layers to ensure that diffusion is not artificially halted, but does not simulate the entire thickness of the wall, to prevent very long run times."**

We chose 2 µm layers because the MLM uses the Euler method in three-dimensions to solve the differential equations. The method will only perform correctly when the spatial step sizes correspond to the typical lengths that the species travel during the chosen time step. We obtained an optimized timestep of 0.01 s, a flow step down the length of the tube of 2 cm, and a diffusion step into the wall of 2 µm. If we decreased these values further, there was minimal effect on the output time series, and a large increase in run time.

R1.3. A result that is implicit through this work is that that nylon delays are similar to PFA, and there is no discussion of outgassing. Should we all just be using nylon for gas sampling, since it is way cheaper? Where is nylon on Figure 8b? Was any outgassing observed? None of this is discussed, and it seems to me could be a radical shift of the atmospheric if I am understanding the results correctly.

We thank the reviewer for bringing up this very important point. Please see R1.1 regarding nylon's irreversible uptake of acids. As acids are species of wide interest in atmospheric chemistry, we do not recommend the community switch to using nylon for all applications. For this reason we also did not include it for further work in the GVS section of the paper (i.e. Fig. 8b). In future work we plan to further explore the implications of nylon's use (e.g. in carpets) on indoor air quality.

R1.4(a) The applications of the GSV are conceptually interesting, but I'm a bit skeptical of its utility. For instance, in the example given using a long length of teflon as a separator (lines 369-374) - couldn't you just sample through a shorter length of chromatography column and get the same effect?

The target applications of the GVS system vs. a gas chromatography column are different. Typical flows through a GC column are a few $cm^3$ $min^{-1}$, while our tubing system can work with 1000 times larger flows. A GC column could not supply enough flow to couple with an OFR or OH reactivity experiment.

We agree that GC will produce better separation for analytical purposes. We do not want to insinuate that polymer tubing could replace proper gas chromatography.

R1.4(b) Similarly, in the application of using the GSV as a volatility pass filter, and in many of the applications of the GVS I can think of, seem to me to include significant complications to interpretation. Because the system is constantly moving toward an equilibrium, the fraction transmitting is constantly changing, so it is not a fixed separator. It is certainly an interesting idea, but strikes me as difficult to draw quantitative meaning from.

We agree that the GVS is a complex system that needs to be used with care. In fact, we have used the GVS in the field in the summer of 2022 as preconditioning to OFR measurements, and found it to be highly useful (Morris, M. A., et al., 2023). When we ran in the field, we only switched tubes every 20 minutes, which was much more practical for the field conditions. We agree that the constant shift toward equilibrium complicates results, but this did not hinder us from gaining important insights into outdoor air quality processes that depend on compound volatility. With the computational models provided, one can certainly account for the changing transmission through the tubes. We understand that this is the first iteration of the design, and others may improve or adapt it where it can be useful.

Morris, M. A., et al. (2023, October 2-6). Measurements of Urban Secondary Organic Aerosol as a Function of Precursor Volatility Class in the Los Angeles Area During Summer 2022 [Platform presentation]. AAAR 2023 Annual Conference, Portland, OR, United States. https://aaarabstracts.com/2023/view_abstract.php?pid=472

Technical comments:
R1.5 Line 109. The definition of delay time is a little confusing, please rephrase

Line 109 originally read:

"We define the partitioning delay time as the time it takes for the tubing output from the start of desorption to reach 10% of the maximum concentration during the absorption period, as is done in previous studies (Pagonis et al., 2017; Deming et al., 2019; Liu et al., 2019)."

and has been replaced with:

**"We define the partitioning delay time as the time it takes to depassivate to 10% of the maximum concentration, as was done in previous studies (Pagonis et al., 2017; Deming et al., 2019; Liu et al., 2019)."**

R1.6 Line 125. The Cole-Parmer 3-way solenoids have a fairly tortured flow path that can introduce delays and losses, even in PFA. Do the authors see a difference in delay times between PFA lines with and without these in line? An issue that has come up in many applications is the impact of these and similar valves on the sample flow stream, this seems like a good opportunity to examine the issue, which would be helpful for the gas sampling community. The authors allude to this on line 356.

To characterize the gas transmission of the 3-way solenoid valves, as suggested in this comment, we have completed an additional experiment. We found that the valves introduced only a minor additional delay for the least volatile species (on top of the PFA tubing delays). We have included the following figure and caption on page 13 of the SI:

[Figure]

[Figure]

[Figure]

**Figure S12. Top panel - time series of the 2-ketones through PFA with no valve installed, PFA with the flow passing through the valve's "normally open" path, and PFA with the flow passing through the valve's "normally closed" path. The normally open flow path results in a room temperature valve, while the normally closed flow path requires power to be opened, resulting in valve heating. Bottom left - a photo of the experimental setup where the valve is not installed. Bottom right - a photo of the valve installed in the "normally open" configuration. With the powered valve installed, delay times did not change compared to just PFA. The room temperature valve contributed ~3 minutes of delay for the $C_{14}$ ketone, 20 seconds of delay for the $C_{13}$ ketone, and no delay change for the rest of the ketones.**

Line 356 originally read:

"We believe this is due to the other components of the setup (e.g. Teflon 3-way valves and fittings) causing more sorption than the components included in the model (tubing and instrument only)."

and has been replaced with:

**"While the 3-way solenoid valves could not be incorporated in the modeling runs, experiments showed that the solenoid valve effects on gas transmission are undetectable for $C_{12}$ and more volatile species, and small for $C_{13}$ and $C_{14}$ ketones (see Figure S12)."**

R1.7 Lines 201-209. I hope you at least automated the brute forcing! If not that sounds exhausting (and more prone to researcher error).

Yes, we automated the model to run the grid of test values on its own. The following text was added to address this point:

Lines 203-205 originally read:

"To do this, we carried out a brute-force fitting routine for each of the 3 parameters, running the model for each combination of parameters, and then calculating the sum squared residual between the experimental and modeled time series. This created a 3D cube of fit errors, which was manually evaluated for each ketone and also across ketones."

and have been replaced with:

**"To do this, we carried out a brute-force method to fit the 3 parameters. The model was automated to run with every combination of input parameters, and then calculate the sum squared residual between the experimental and modeled time series. This created a 3D cube of fit errors, which was manually evaluated for each ketone and also across ketones."**

R1.8. Table 2. Values for Cw here are an order of magnitude larger for the same materials tested by Deming et al. Is that a function just of different tubing geometries, and if so, can they be made more directly comparable? If not, what is the reason for that?

This highlights an important point, which we have addressed in the manuscript with the following text in line 277:

**"There are some differences between the values of $C_w$ determined in our experiments and those of previous studies. Deming et al. report $C_w$ values about an order of magnitude smaller than those determined here. The Deming et al. experiments were conducted with a quadrupole PTRMS that had a much longer response time than the Vocus PTRMS used in our work. The quadrupole response time was subtracted from the tubing time, and this procedure may have introduced some inaccuracy in those results. The flow rate was also ~10 times lower in the Deming et al. work, although that should not affect the results in principle. Liu et al. report $C_w$ values much lower than expected when extrapolated to the ketone volatility range. The main reason for this difference is that $C_w$ has been shown to decrease strongly as $C^*$ decreases (Liu et al., 2019; Krechmer et al., 2020; Huang et al., 2018). This effect is thought to be due to increasing activity coefficients in non-polar Teflon for the more polar multifunctional species used by Liu et al., compared to the less polar ketones used in our work. We recommend that users of this method constrain the $C_w$ values for their tubing by fitting their data with the models we provide."**

R1.9 Line 291. Typo, should be "cPFA"

In line 291, **"cPTFA"** has been changed to **"cPFA"**

R1.10 Figure 7. It does not look obvious to me that silconert has a much slower response time than cPFA, though Figure 5 (and the text) says so. Am I just unable to see it clearly, or am I missing something. Also, the darkest blue line seems to be missing from cPFA during desorption, or maybe the colors are just different.

We agree that the absorption and desorption timescales of cPFA and SilcoNert are similar in Fig. 7. We had already included the lengths of tubing used in the caption, because they were not the same (149 cm for cPFA vs. 61 cm for the stainless steel tubes). When normalized to the tubing length, cPFA significantly outperforms SilcoNert.

We have replaced Figure 7 (updated figure shown below) to fix the coloring issue (the blue background had come forward, distorting the colors; the data did not change).

[Figure]

R1.11 Line 324. Was cPFA or cPTFE similarly tested for losses? Perhaps that was shown in a previous publication? That would be helpful, since a conclusion of Figure 5 is that cPFA is a better option than the coated metal options (and easier to work with).

The following text was added to line 286:

**"Deming et. al. tested cPFA tubing for particle losses, and found them to be comparable to copper tubing."**

R1.12 Line 332-336. It is somewhat unorthodox to include in the SI a whole subject/experiment.discussion that is not really a part of the main work or main text. I understand that these experiments may not constitute a separate paper, so perhaps this is a appropriate, but it is certainly unusual, and I would encourage the authors to consider whether this manuscript is actually the right place for this information given that the whole section is relegated to the SI and no other discussion or use of OFRs is included in the main text.

We thank the reviewer for bringing up this point. We understand that it is unusual to include a section like this in the SI. However, the subset of our community that uses OFRs will find this information helpful. So that others don't have to repeat our tests, we would like to keep this section in the SI, or otherwise it would remain unpublished and be lost over time. We feel that it conceptually fits with this manuscript, as we use the methods of this manuscript to do the testing, and we do not feel that it contains enough material to constitute a technical paper of its own.

R1.13 Line 335. "Supplementary information" or something like that instead of "supplemental"

In line 335, **"supplemental"** has been changed to **"supplementary information"**

R1.14 Line 343-345. There is no counterfactual shown about what happens when a PFA period does not follow cSI - does it increase less quickly? The following statement about depassivation is reasonable, but there is no real evidence presented to demonstrate it.

We have clarified this text to reduce confusion. Although an additional experiment could be done, this appears trivial to us given the data already presented in the paper.

Lines 344-347 originally read:

"When a PFA sampling period follows a cSI sampling period, the ketone concentrations rise quickly before reaching their expected transmission levels. This is because the cSI tubing is so absorptive that the shared PFA lines downstream of the GVS are depassivated while the cSI tubing is sampled, and when the system switches over to PFA from cSI, those lines reabsorb the ketones similar to the start of the experiment."

and have been replaced with:

**"The system has a shared outlet (~ 1.5 m PFA tube) that connects to the mass spectrometer. When the inlet material is a highly sorptive material like cSI, the shared outlet is partially depassivated, and then requires time to be passivated again when the system switches to using PFA as the inlet material. This is seen in the time series, as every time the inlet material is PFA, there is a steeper transient in the concentrations of the species."**

R1.15 Line 388-392. It is beyond the scope of this work for sure, but I'm curious if the authors have any thoughts on how to incorporate these findings into a better understanding of indoor air? How would one go about modeling a carpet, which has very different geometry and surface area ratios than a tube? Painted surfaces are fairly uniform, but many of the surfaces described here have complex fractal-like geometries.

This is a great question. Our ongoing work includes rolling carpet up like a tube, to adapt the methodology in this manuscript to that more complex problem. We have plans to quantify the "unmoving/dead air" inside the complex surface of carpet using $CO_2$, to try and simplify the modeling. Certainly this will be a challenging exploration; we plan to build on the work other researchers have done already to model the air-carpet interface. (Pei et al., 2022)

Pei, G., Xuan, Y., Morrison, G., and Rim, D.: Understanding Ozone Transport and Deposition within Indoor Surface Boundary Layers, Environ. Sci. Technol., 56, 7820–7829, https://doi.org/10.1021/acs.est.1c08040, 2022.

In addition to the changes listed above, we have made some additional small changes for clarity, described below. These small changes do not impact any of the conclusions of the paper.

1. We have included more specifications for the oxidation flow reactor coating. To page 14 in the SI, we have added the following text:

**"One was uncoated (aluminum with chromate finish) and the other was coated (aluminum with chromate finish, coated with a Chemours 2-part conductive PTFE coating, 855G-021 primer and 855G-103 topcoat). The coated OFR is from the 2020-2022 coating batch. "**

2. We have included gas transmission data on another material, a halocarbon wax coated glass tube. We have included manufacturing details for the material in table S2:

| Halocarbon Wax (polychlorotrifluoroethylene) Coated Quartz | 1.05 | 61 | Halocarbon Products Corporation (2300 Series) | - |
|---|---|---|---|---|

We updated Fig. 5 to include this material, reposted below:

[Figure]

Finally, we have added an acknowledgement to **"David Osborn of Sandia National Labs for providing the halocarbon-wax coated glass tube and useful discussions"**

---

## Author Comment (AC2)

**Response to reviewers for the paper "Absorption of VOCs by polymer tubing: implications for indoor air and use as a simple gas-phase volatility separation technique" Melissa A. Morris , Demetrios Pagonis , Douglas A. Day , Joost A. de Gouw , Paul J. Ziemann , and Jose L. Jimenez**

We thank the reviewers for their comments on our paper. To guide the review process we have copied the reviewer comments in black text. Our responses are in regular blue font. We have responded to all the referee comments and made alterations to our paper **in bold text**. All line numbers refer to the originally submitted manuscript.

**Anonymous Referee #2**
Overview:
R2.0. The manuscript describes a new measurement technique to separate gas-phase organic compounds by volatility using different types of sampling tubing. This is an innovative approach that can be leveraged by the broader atmospheric research community to address major research challenges related to, for example, gas and aerosol chemistry of volatile organics in complex mixtures. It is also incredibly valuable that the research group publishes all the models they have developed on their website that is open access to everyone. The introduction is very clear and nicely places the work in context with prior literature. The figures are publication quality and clearly demonstrate major findings from the systematic testing of partitioning delays of different tubing material across a range of compound volatilities. However, the paper seems to end prematurely without demonstrating the use of the gas volatility separator technique in a real measurement application with a mixture of organics. How would one leverage this technique to develop plots of the relative contribution of different volatility classes in the original sample? That still isn't entirely clear. I also think the manuscript could be strengthened by focusing the tables and figures in the main manuscript on the most critical findings and leaving some of the details for the SI. It was a little challenging to pull the main points out of all the details. I recommend the paper for publication after some (mostly) minor revisions that address the concerns summarized here and described below.

Major comments:
R2.1 One of the key findings is the tubing categorization as slow vs fast vs adsorption only. Figure 2 is a nice illustration of the differences between the tubing types for the range of compound volatilities studied. Figure 3 is more challenging to read. Particularly 3b. There are too many lines with too many different colors and dashes. If the main point is just to illustrate that the SLM does not work for polypropylene, I think it would be more effective to show the time-series example for 1 of the ketones (not 2) and then also provide a more quantitative metric for the model prediction for all the compounds in a table next to the graph. What was your metric for deciding when the SML model worked or it didn't? You provide all the details about the fit parameters for the model but no clear quantitative description to evaluate how well the models did at predicting the measurements. Even just a simple linear regression comparing modeled vs measured values at each time point could possibly work to draw a more objective line between a "good fit" and a "poor fit."

For figure 3, we think that it is important to show both $C_{10}$ and $C_{12}$ ketones for the single vs multiple layer model in panel (b), as it illustrates some key features of the model and the data. However, we understand that having so many traces in one graph can be confusing. We have revised the graph to make 4 instead of

2 panels for increased clarity. The revised figure is reproduced below:

[Figure]

Please also see the response to R1.2(c), where we have made changes to the manuscript to be more quantitative with our model fitting metrics.

R2.2 Section 3.1.3 doesn't appear to be providing critical information about the key results. The only figures cited are in the SI, not in the main text. Perhaps this entire section could be moved to the SI? It reads as a minor technical point.

We believe that this short section adds value in the main manuscript. It explains why the model fits are still imperfect with the multilayer model, and points the reader to the literature that would need to be explored if a better fitting model was desired.

R2.3a Section 3.2. It was surprising to read that the absorption delay times are often larger than the desorption delay times for the slow-diffusion materials. This point appears to be glossed over fairly quickly, but could the authors describe the implications of this for using the GVS technique to classify volatility distributions from an ambient measurement?

As discussed in section 3.1.3 and our previous response to R2.2, the material-VOC interactions show some more complex behaviors that are not perfectly captured by the model. As also discussed in response to comment R1.8 (and the text added to the manuscript quoted there), it is important for users of this method to perform their own calibrations with the tubing system and species of interest.

In our field application mentioned above (response to R1.4b) we used a Vocus PTRMS to sample the output of the tubing, which allows a clearer interpretation of the data.

R2.3b How long does slow-diffusion tubing need to be conditioned before reaching equilibrium?

The timescale to reach equilibration when sampling through the slow-diffusion tubing depends on the

flow rate, the geometry of the tubing, and the volatility of the compounds sampled. In our work, at a nominal flow rate of 2 LPM, slow-diffusion tubing reached equilibrium with highly volatile species in 1 second or less, while other species took up to 25 minutes to reach equilibrium. As this is apparent from the time series shown in the figures, we have not modified the manuscript in response to this point.

R2.3c Why would there be more variability in absorption delays than desorption delays?

See response to R2.3a.

R2.4 Does it make sense for Section 3.2 to be its own section? I think the description of key findings in Figure 5 would be more clear if they were discussed in the same section instead of separated between sections 3.2 and 3.3.1 and 3.3.2.

We prefer to keep the sections as they are. The logic for their separation is clear, as 3.2 refers to materials used to sample gases only, while 3.3 discusses materials used to sample gases and particles simultaneously. This concerns two mostly separate research communities, and thus we think the presentation of the material in this order is quite practical.

R2.5 Figure 8b is the most important figure of the entire paper and it is buried at the end. I think it would be more effective to move this before some of the technical details about humidity effects on desorption of metal tubing or contaminant peaks in conductive tubing (new vs aged).

We also like Fig. 8b a lot, but we do not think it is "buried." It needs to be discussed after the conductive silicone tubing has been discussed. It makes most sense to us to discuss the silicone tubing artifacts in the same section as all other aspects of that tubing. Also, the section discussing the stainless tubings is conceptually similar to the preceding sections, and more different from the final section on the GVS. Thus we prefer to keep the current order of the sections, with Figure 8b at the end, as it relies on information shared throughout the manuscript.

R2.6 The abstract states, "we demonstrate how to use a combination of slow- and fast-diffusion tubing to separate a mixture of VOCs into volatility classes," but the paper actually stops just short of actually demonstrating that. Can you take the data shown in Figure 8a and develop a synthesized volatility distribution of the starting mixture from that data? Figure 8b shows the distribution being transmitted through each of the different tubes, but the demonstration is missing that final step of actually reverse-engineering the original volatility distribution. I think it would have been even more compelling to demonstrate how this would work with two different mixtures composed of contrasting contributions from SVOC/IVOC compounds.

We understand the reviewer's point. However, it would be circular to use the data for the ketones to determine the properties of the GVS, and then use the GVS to retrieve the volatility distribution of the ketones. We do not have the system set up at present to conduct more experiments with other mixtures, which in any case would be a substantial amount of work, perhaps enough for its own separate manuscript. Instead, we have changed the sentence in the abstract to read:

**"We demonstrate a system combining several slow- and fast-diffusion tubings that can be used to separate a mixture of VOCs into volatility classes."**

Technical comments:
R2.7 Lines 252-255: It wasn't entirely clear what the authors meant by "short times" and "long times" in this description. Do you mean earlier in the absorption phase and later in the absorption phase of the sampling cycle?

We have reworded this text to clarify this point. Lines 250-256 originally read:

"The desorption of the fast diffusion materials proceeded at a changing rate, initially being faster than the corresponding absorption rate, and later being slower than the corresponding absorption rate (as shown in Fig. S7). This meant that, at short times, compounds diffused out of the polymer faster than they went in, and at long times, compounds diffused out of the polymer slower than they went in. The behavior at long times is expected; to enter the polymer, compounds just partition to the surface, but to exit the polymer, compounds must diffuse back to the surface before re-partitioning into the gas phase, therefore taking longer to exit than enter. The behavior at short times was a surprise."

and have been replaced with:

**"The desorption of the fast diffusion materials proceeded at a changing rate, initially being faster than the corresponding absorption rate, and later being slower than the corresponding absorption rate (as shown in Fig. S7). This meant that, early in the absorption period, compounds diffused out of the polymer faster than they went in, and later in the absorption period, compounds diffused out of the polymer slower than they went in. The behavior at later times is expected; to enter the polymer, compounds just partition to the surface, but to exit the polymer, compounds must diffuse back to the surface before re-partitioning into the gas phase, therefore taking longer to exit than enter. The behavior early in the absorption period was a surprise."**

R2.8 Line 260: unclear what is meant by having to "split the difference" to model the whole time series.

Lines 259-261 originally read:

"We noticed this anomaly in our data while modeling; the multilayer model could reproduce either the absorption period or the desorption period for a material very well, but had to split the difference between the two periods to model the whole time series (as shown in Fig. S8)."

and have been replaced with the following text to clarify this point:

**"We noticed this anomaly in our data while modeling. When the model was asked to optimize the fit parameters to just the absorption period of the data, the resulting desorption period had high model error (as shown in Fig. S8). The inverse was true when the model was asked to optimize the fit parameters to just the desorption period. When the model was asked to optimize the fit parameters to the whole time series, they fell in between the absorption-only and desorption-only fit**

**parameters, and provided an adequate but not perfect fit to both periods."**

R2.9 Lines 329-331: I think this sentence is supposed to be the final sentence of the preceding paragraph. Otherwise it is unclear what "these" materials refers to.

Lines 328 and 329 have become one paragraph.

R2.10 Line 355: typo in the sentence, "We believe this due to the…"

Line 355 has been revised to read "We believe this **is** due to the…"
* * *
In addition to the changes listed above, we have made some additional small changes for clarity, described at the end of the response to reviewer #1.

---

## Author Comment (AC3)

**Response to Andrew Whitehill**

To guide the commenting process we have copied the comments in black text. Our responses are in regular blue font, with alterations to our paper **in bold text**. All line numbers refer to the originally submitted manuscript.

Overview:
This is an excellent manuscript and a worthwhile contribution to our continued understanding of gas-polymer interactions, especially for the choice of sample tubing for different purposes. I have a few minor comments that I think would improve the paper. I appreciate the excellent work done here and hope to see this contribution published.

Major Comments:
R3.1 Cole-Parmer Part No. EW-01540-18 is advertised as a PTFE-body solenoid valve. The manuscript claims it is PFA. Given the minor differences observed in the behavior of PTFE vs PFA this differentiation could be important.

The authors would like to thank the reviewer for catching this error. The valves are indeed made of PTFE, not PFA.

Lines 125 and 126 have been revised to **"PTFE 3-way solenoid valve"**

R3.2 Please specify the type of stainless steel tubing used. Although specific manufacturers / part numbers are provided for most of the polymer tubing, no specific information is provided about the stainless steel tubing. Different types of stainless steel (e.g. 304 vs 316) and surface finishes would likely make a significant difference on transmission of specific compounds through the tubing.

This is an interesting point. Unfortunately we do not have the surface finish specifications of these tubes to report. Since the steel tubes used in this work (except for the reference tube, made from the same cut of metal) undergo a coating process, we assume that the original surface finish of the steel does not affect the absorptive capacity of the final coated product.

The paper by Deming et. al. has a longer discussion of metal tubing and may be of interest. They showed that gas transmission through steel tubes is highly dependent on the humidity and concentration of gases present, as well as their sampling history, and is subject to such complexity that we do not recommend ever using metal tubing to separate gas mixtures.

R3.3 All the compounds tested are 2-ketones. Although these are interesting and important compounds it is not clear how well these results will extrapolate to other compounds or compound classes. The implications of testing 2-ketones and how the results apply to non 2-ketones should be discussed a bit more if not illustrated with non 2-ketone experiments.

The authors are thankful for this important point. Please see R1.1 for discussion of this point and the text added to the manuscript.